# Optimal Route Planning for Truck–Drone Delivery Using Variable Neighborhood Tabu Search Algorithm

**Bao Tong [1], Jianwei Wang [1,2,3,4,*], Xue Wang [2,3,4,*], Feihao Zhou [1,2,3,4], Xinhua Mao [1,2,3,4] and Wenlong Zheng [2,3,4]**

1 College of Transportation Engineering, Chang'an University, Xi'an 710064, China; 13521256788@163.com (B.T.); zhoufeihao66@163.com (F.Z.); maoxinhua@chd.edu.cn (X.M.)
2 Engineering Research Center of Highway Infrastructure Digitalization, Ministry of Education, Xi'an 710064, China; zhwenlong@chd.edu.cn
3 Engineering Research Center of Digital Construction and Management for Transportation Infrastructure of Shaanxi Province, Xi'an 710064, China
4 Xi'an Key Laboratory of Digitalization of Transportation Infrastructure Construction and Management, Xi'an 710064, China
* Correspondence: wjianwei@chd.edu.cn (J.W.); wangxue@chd.edu.cn (X.W.)

**Abstract:** The optimal delivery route problem for truck–drone delivery is defined as a traveling salesman problem with drone (TSP-D), which has been studied in a wide range of previous literature. However, most of the existing studies ignore truck waiting time at rendezvous points. To fill this gap, this paper builds a mixed integer nonlinear programming model subject to time constraints and route constraints, aiming to minimize the total delivery time. Since the TSP-D is non-deterministic polynomial-time hard (NP-hard), the proposed model is solved by the variable neighborhood tabu search algorithm, where the neighborhood structure is changed by point exchange and link exchange to expand the tabu search range. A delivery network with 1 warehouse and 23 customer points are employed as a case study to verify the effectiveness of the model and algorithm. The 23 customer points are visited by three truck–drones. The results indicate that truck–drone delivery can effectively reduce the total delivery time by 20.1% compared with traditional pure-truck delivery. Sensitivity analysis of different parameters shows that increasing the number of truck–drones can effectively save the total delivery time, but gradually reduce the marginal benefits. Only increasing either the truck speed or drone speed can reduce the total delivery time, but not to the greatest extent. Bilateral increase of truck speed and drone speed can minimize the delivery time. It can clearly be seen that the proposed method can effectively optimize the truck–drone delivery route and improve the delivery efficiency.

**Keywords:** optimal route problem; truck–drone delivery; mixed-integer nonlinear programming model; variable neighborhood tabu search algorithm

## 1. Introduction

The new concept of Logistics 4.0 requires higher quality standards of demand, distribution, and inventory management [1]. Meanwhile, the prosperity of e-commerce and mobile business requires more efficient logistics [2]. With the rapid development of drone technology, drones have been widely applied in health care, agriculture, aerial photography, mapping, etc. In the last few years, some e-commerce and logistics companies started to adopt drones in deliveries, which greatly reduces delivery costs and improves delivery efficiency. DHL company reported in 2015 that it successfully delivered medicals and other goods considered as urgent on a small island in northern Germany using drones. Amazon carried out its first drone delivery practice in 2017. However, drone-only delivery has limitations in delivery coverage and carrying capacity due to the drones' endurance. Hence, the coordinated delivery of trucks and drones (or truck–drone delivery) attained



considerable attention [3]. Accordingly, there is a growing demand for developing an efficient route planning method for the truck–drone delivery problem [4].

In the literature, many studies focus on the route planning for truck–drone delivery of goods from warehouses to customers, which is defined as a traveling salesman problem with drone (TSP-D) [5]. In summary, according to the functions of drones in the delivery process, the TSP-D can be classified into four categories: (1) flying sidekick traveling salesman problem (FSTSP) [6,7], (2) parallel drone-scheduling traveling salesman problem (PDSTSP) [8–10], (3) heterogeneous delivery problem (HDP) [11–13], and (4) vehicle routing problem with drone resupply (VRPDR) [14]. This study mainly studies FSTSP. The existing FSTSP assumes that the drones arrive at rendezvous points ahead of the trucks. But the drones may arrive at rendezvous points later than the trucks, which would result in truck waiting time.

To this end, this paper aims to investigate the optimal route planning for truck–drone delivery, considering truck waiting time at rendezvous points. Our work makes the following three contributions. (1) We build an optimal route planning model with time constraints and route constraints, aiming at the minimization of the total delivery time. (2) We design a variable neighborhood tabu search algorithm as the model solution. (3) We demonstrate the proposed mathematical model and algorithm on a delivery network and analyze the effects of model parameters on the delivery efficiency.

The remainder of this paper is organized as follows. Section 2 reviews the previous studies on TSP-D and the solution approaches. Section 3 develops a mathematical model to formulate the optimal route planning for truck–drone delivery. Section 4 presents the model solution. Section 5 utilizes a case study to test the proposed method framework and analyze the sensitivity of model parameters. Conclusions and future work are discussed in Section 6.

## 2. Literature Review

The existing studies of TSP-D includes: FSTSP, PDSTSP, HDP, and VRPDR. FSTSP means that a truck carries one or multiple drones, each drone and the truck deliver goods or parcels together, and when each drone completes a delivery task, it returns to the truck to load goods and perform the next delivery task [15]. PDSTSP indicates that both drones and trucks start from a warehouse and deliver goods independently; when each drone completes a delivery task, it goes back to the warehouse to load goods and deliver them to the next customer [8]. HDP represents a scenario in which drones carry out all delivery tasks, while trucks mainly carry drones to specific locations and serve as moving landing points and take-off points for drones [16]. The definition of VRPDR is that all goods are delivered to customers by trucks, and drones are used to resupply goods to trucks from the warehouse [14].

The optimal route planning problem for truck–drone delivery in this paper occurs with the FSTSP, which is an extension of the traveling salesman problem first proposed by Murray and Chu in 2015 [15]. Due to complexity, Murray and Chu assume that there is only one truck and one drone in the FSTSP. Based on the common assumptions of the FSTSP, Agatz et al. put forward a derived FSTSP, which defines that each drone's launch point and return point are in the same position, and the delivery route is affected by the flying distance instead of flying time [17].

Optimization methods are commonly used to study the FSTSP in the literature. Sacramento et al. develop the FSTSP with multiple trucks and multiple drones, which is formulated as a mixed-integer programming model with the objective of minimum delivery costs [18]. Ham establishes an optimal route planning model considering a delivery time window, and utilizes the constrained programming method to solve the problem [8]. Based on the Bellman–Held–Karp dynamic programming approach, Bouman et al. employed a three-step exact solution method to solve the FSTSP [19]. The three-step exact solution method can help significantly reduce the solution times, while having impact on the overall solution quality.

Since it has been proven that the FSTSP is non-deterministic polynomial-time hard (NP-hard), heuristic algorithms are increasingly used to solve this problem. Schermer et al. use a variable neighborhood search hybrid heuristic algorithm [20] and Dorling et al. apply a simulated annealing algorithm to solve the FSTSP [21]. Ha et al. develop two heuristic algorithms, i.e., greedy random adaptive search algorithm (GRASA) and traveling salesman problem local search algorithm (TSP-LSA) [22]. GRASA is a metaheuristic algorithm, which first generates a traveling salesman problem for trucks, then uses a decomposition algorithm to remove some customers from the trucks' delivery route and assigns these customers to the drones. TSP-LSA is modified from the reassign heuristic proposed by Murray and Chu [15]. Compared with the reassign heuristic, the TSP-LSA has differences in calculating delivery cost savings and assigning customers between trucks' delivery routes and drones' delivery routes. GRASA has higher solution quality than TSP-LSA, while it needs more computing time. Yurek and Ozmutlu design a two-stage decomposition heuristic algorithm [23]. In the first stage, a constructive greedy heuristic is utilized to assign the customers to the drones' delivery routes and the trucks' delivery routes. In the second stage, a nonlinear programming problem is solved to obtain the drones' delivery routes, which can minimize the drones' waiting time at rendezvous points. Gonzalez-R et al. extend the FSTSP by enabling drones to deliver goods to more than one customer between two rendezvous points, and solve the extended FSTSP using a simulated annealing algorithm so as to get the globally optimal solution [7].

Despite the wide range of studies in FSTSP, there are still some gaps in previous studies to be filled. Different from previous studies, which commonly assume that the drones arrive at rendezvous points ahead of the trucks, this research takes truck waiting time at rendezvous points into account under the scenario where the drones arrive at rendezvous points later than the trucks. In addition, the efficiency of the neighborhood search approach in most of these studies to obtain the global optimal solution for this large-scale combinatorial optimization problem should be improved.

To address the above issues, we formulate and solve the optimal route planning for truck–drone delivery considering truck waiting time at rendezvous points. In the model solution, a three-stage method is proposed to generate the initial solution, and a variable neighborhood tabu search algorithm is developed.

## 3. Methodology

### 3.1. Assumption

This research is carried out based on the following assumptions:

(1) Drones will not fail during the entire delivery process.
(2) Each drone can only execute one delivery task after each take-off.
(3) Each customer is visited only once by a drone or a truck.
(4) All trucks have the same service time at each customer point, and all drones also have the same service time at each customer point.

### 3.2. Notation and Problem Statement

It is assumed that at the initial time $u_0$, a fleet $N = \{1, 2, \cdots, v\}$ consisting of $v$ trucks needs to deliver $m$ parcels to $m$ different customers $C = \{1, 2, \cdots, m\}$. One customer receives one parcel, and each truck is equipped with a drone. Trucks equipped with drones depart from the warehouse and deliver parcels to customers one by one, following a given route. All customers can be served by a truck or a drone. After the delivery is completed, trucks and drones return to the warehouse. For simplicity, the warehouse is represented by $D = \{0, m+1\}$, that is, the truck departs from $\{0\}$ and returns to $\{m+1\}$. Therefore, the set of all nodes in the delivery network is $E = \{0, 1, 2, \cdots, m, m+1\}$, and the truck can start from the set of nodes $E^+ = \{0, 1, 2, \cdots, m\}$ and reach the set of nodes $E^- = \{0, 1, 2, \cdots, m, m+1\}$. The set of delivery routes in the delivery network is $A = \{(i, j): i \in E^+, j \in E^-, i \neq j\}$. Figure 1 shows the truck–drone delivery routes.

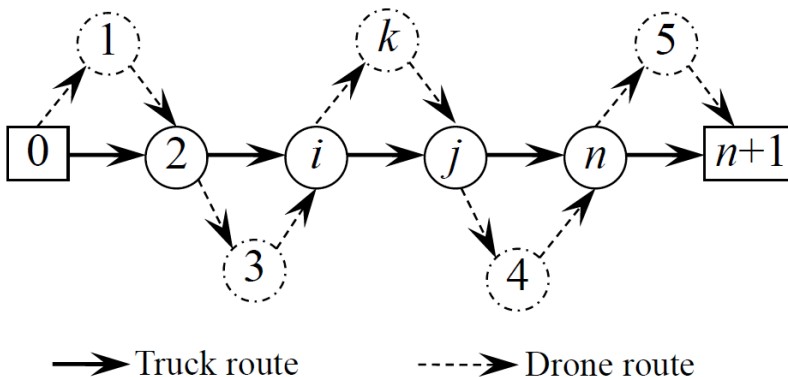

**Figure 1.** Truck–drone delivery routes.

Drones can only take off on nodes $E^+ = \{0, 1, 2, \cdots, m\}$ and can only land on nodes $E^- = \{0, 1, 2, \cdots, m, m+1\}$. A truck delivers a parcel at customer point $i$ and launches the drone at $i$, then the truck goes to the next customer point $j$, and the drone goes to customer point $k$. After completing the delivery task, the drone returns to customer point $j$ and is collected by the truck. Swap the drone battery with a new or fully recharged battery and load the drone with the parcel for the next delivery task at customer point $j$. Meanwhile, the truck continues to the next customer point until all parcels are delivered, then returns to the warehouse.

Each truck route can be represented by two nodes, i.e., $(i, j): i \in E^+, j \in E^-, i \neq j$. Different from the truck route, each drone route is represented by three nodes, i.e., $(i, k, j)$, which means that a drone takes off at point $i$, $i \in E^+$, delivers a parcel to point $k$, $k \in \{E^-: k \neq i\}$, and returns to point $j$, $j \in \left\{C: j \neq i, j \neq k, t_L + t_R + t_{ik}^D + t_{kj}^D + s^D \leq e\right\}$. $t_L$ and $t_R$ are the time required for a drone to take off and land, $t_{ik}^D$ is the time it takes for a drone to fly from launch point $i$ to customer point $k$, and $t_{kj}^D$ is the travel time of a drone from customer point $k$ to land point $j$. $s^D$ is the service time of a drone at a customer point, and $e$ is the maximum endurance of a drone. We define the set of all drone routes as $P = \{(i, k, j)\}$, the set of drone routes starting from point $i$ as $P_i^+$, and the set of drone routes ending at point $j$ as $P_j^-$.

Therefore, the problem studied in this paper is defined as: when the number of trucks (one truck carries one drone) is given, determine the truck–drone delivery routes as well as the sequence of each delivery task, so as to minimize the total delivery time.

To this end, we define the following decision variables:

$x_{ij}^n$: If truck $n \in N$ goes from point $i \in E^+$ to point $j \in E^-, j \neq i$, $x_{ij}^n = 1$; otherwise $x_{ij}^n = 0$;

$y_p^n$: If the delivery route of the drone on truck $n \in N$ is $p \in P$, $y_p^n = 1$; otherwise $y_p^n = 0$;

$u_j^n$: The time when truck $n \in N$ arrives at customer point $j \in E^-$;

$u_j^{n\prime}$: The time when the drone on truck $n \in N$ arrives at customer point $j \in E^-$;

$z_i^n$: The position of node $i \in E$ on the route of truck $n \in N$.

### 3.3. Formulation of the Problem

Based on the above assumptions, we formulate the proposed problem as a single objective mixed-integer nonlinear programming model.

Objective function:

$$\min \tau = \max_{n \in N}\left\{u_{m+1}^n, u_{m+1}^{n\prime}\right\} - u_0 \tag{1}$$

Constraint conditions:

$$\sum_{i \in C} x_{0i}^n = \sum_{i \in C} x_{i,m+1}^n = 1, \forall n \in N \tag{2}$$

$$x_{0,m+1}^n = 0, \forall n \in N \tag{3}$$

$$\sum_{n \in N} \sum_{i \in C} x_{0i}^n = \sum_{n \in N} \sum_{i \in C} x_{i,m+1}^n = v \tag{4}$$

$$\sum_{j \in E^-, i \neq j} x_{ij}^n = \sum_{j \in E^+, i \neq j} x_{ji}^n, \forall n \in N, \forall i \in E \tag{5}$$

$$\sum_{n \in N} \sum_{i \in E^+, i \neq j} x_{ij}^n + \sum_{n \in N} \sum_{p \in P} y_p^n = 1, \forall j \in C \tag{6}$$

$$\sum_{p \in P_i^+} y_p^n = \{0, 1\}, \forall i \in E^+, \forall n \in N \tag{7}$$

$$\sum_{p \in P_j^-} y_p^n = \{0, 1\}, \forall j \in E^-, \forall n \in N \tag{8}$$

$$\sum_{p \in P_i^+} \sum_{i \in E^+} y_p^n = \sum_{p \in P_j^-} \sum_{j \in E^-} y_p^n, \forall n \in N \tag{9}$$

$$2y_p^n \leq \sum_{\forall h \in E^+, h \neq i} x_{hi} + \sum_{l \in C, l \neq j} x_{lj}, \forall i \in C, \forall n \in N, \forall p \in P \tag{10}$$

$$y_p^n \leq \sum_{\forall h \in E^+, h \neq j} x_{hj}, \forall k \in C, \forall n \in N, \ p = (0, k, j) \in P \tag{11}$$

$$u_0^{n'} = u_0^n = u_0, \forall n \in N \tag{12}$$

$$z_i^n + 1 \leq z_j^n + M\left(1 - x_{ij}^n\right), \forall n \in N, \forall i \in E^+, \forall j \in E^- \tag{13}$$

$$z_j^n \leq M \sum_{i \in E^+} x_{ij}^n, \forall n \in N, \forall j \in E^- \tag{14}$$

$$u_j^{n'} - u_i^{n'} \leq e + M\left(1 - y_p^n\right), \forall p \in P, \forall n \in N \tag{15}$$

$$u_i^n + \Delta u_i^n + t_{ij}^T \leq u_j^n + M\left(1 - x_{ij}^n\right), \forall n \in N, \forall (i, j) \in A \tag{16}$$

$$\Delta u_i^n = s^T + t_L \sum_{p \in P_i^+} y_p^n + t_R \sum_{p \in P_j^-} y_p^n + \pi_i^n, \forall n \in N, \forall i \in E^+ \tag{17}$$

$$\pi_i^n = \max\left\{0, u_i^{n'} - u_i^n - s^T\right\}, \forall n \in N, \forall i \in E^+ \tag{18}$$

$$u_i^n + t_{ik}^D + t_L \leq u_k^{n'} + M\left(1 - \sum_{p \in P} y_p^n\right), \forall n \in N, \forall (i, k) \in A \tag{19}$$

$$u_k^{n'} + t_{kj}^D + s^D + t_R \leq u_j^{n'} + M\left(1 - \sum_{p \in P} y_p^n\right), \forall n \in N, \forall (k, j) \in A \tag{20}$$

$$t_{ij}^T = L_{ij}/V^T, \forall (i, j) \in A \tag{21}$$

$$t_{ij}^D = L_{ij}/V^D, \forall (i, j) \in A \tag{22}$$

$$x_{ij}^n \in \{0, 1\}, \forall i \in E^+, \forall j \in \{E^- : j \neq i\}, \forall n \in N \tag{23}$$

$$y_p^n \in \{0, 1\}, \forall p \in P, \forall n \in N \tag{24}$$

$$u_j^n, u_i^{n'} \geq u_0, \forall j \in E^-, \forall n \in N \tag{25}$$

$$z_i^n \geq 0, \forall i \in E, \forall n \in N \tag{26}$$

where $\tau$ is the total delivery time; $M$ is a sufficiently large number; $h$ and $l$ are the customer points different from $i$ and $j$; $\Delta u_i^n$ is the time that the truck $n$ spends at the customer point $i$;

$t_{ij}^{\mathrm{T}}$ is the travel time required by the truck from $i$ to $j$; $t_{ij}^{\mathrm{D}}$ is the travel time required by the drone from $i$ to $j$; $s^{\mathrm{T}}$ is the service time of the truck at the customer point; $\pi_i^n$ is the waiting time of truck $n$ at the customer point $i$; $L_{ij}$ is the length of route $(i, j)$; $V^{\mathrm{T}}$ is the average driving speed of the trucks; $V^{\mathrm{D}}$ is the average flying speed of the drones.

Equation (1) is the objective function of the model, which seeks to minimize the total delivery time. The time consumption of the entire delivery process is the arrival time of the last truck or drone returning the warehouse minus the initial departure time. Equation (2) indicates that each truck starts from the warehouse and finally returns to the warehouse, and only departs once and returns once. Equation (3) indicates that no truck will return to the warehouse immediately after departure. Equation (4) indicates that $v$ trucks jointly complete the entire delivery process. Equation (5) is the capacity constraint of the truck. Equation (6) indicates that each customer is either visited by a truck or by a drone, and is only visited once. In Equation (7), each drone either does not take off or only takes off once at any possible launch point. Equation (8) ensures that each drone either does not land or only lands once at any possible landing point. Equation (9) is the capacity constraint of the drone. In Equation (10), if the drone launches from customer point $i$ and is collected by the truck at customer point $k$, the truck must visit customer point $k$. Equation (11) states that if the drone launches from the warehouse 0 and is collected by the truck at customer point $j$, the truck must visit both customer points $i$ and $j$. Equation (12) means that both drones and trucks start from the warehouse at initial time $u_0$. Equations (13) and (14) are the subtour elimination constraints for the truck, referring to the position of customers that are visited on the truck's route. Equation (15) means that the time of each delivery by drone cannot exceed the maximum endurance of the drone battery. Equation (16) ensures that the sum of the time the truck arrives at customer point $i$ plus the time the truck spends at $i$ and the time it travels to customer point $j$ is less than or equal to the time the truck arrives at customer point $j$. Equation (17) means that the time that the truck spends at customer point $i$ is the sum of the truck's delivery time at $i$ plus the time required for the drone to launch (or land) at customer point $i$ and the truck's waiting time at customer point $i$. Equation (18) calculates the truck's waiting time at a customer point. In Equation (19), the time when the drone reaches customer point $k$ is greater than or equal to the time when the truck reaches the drone's launch point $i$ plus the time required for the drone to take off and the drone's travel time from $i$ to $k$. Equation (20) means that the time when the drone reaches its landing point (customer point $j$) is greater than or equal to the time when the truck reaches customer point $j$ plus the drone's delivery time at customer point $k$, the time it takes for the drone to land at customer point $j$, and the drone's travel time from $k$ to $j$. Equation (21) calculates the truck's travel time. Equation (22) calculates the drone's travel time. Equations (23)–(26) define the types of decision variables.

## 4. Model Solution

The optimal route problem for truck–drone delivery is NP-hard [24], and the tabu search algorithm is a commonly used metaheuristic algorithm to solve NP-hard problems [25]. The traditional tabu search algorithm generates a fixed neighborhood by searching, which affects the search range and reduces the convergence speed [26]. Considering that the real delivery process covers many customer points, the traditional tabu search algorithm has limitations in solving this large-scale combinatorial optimization problem. Therefore, we use a variable neighborhood tabu search algorithm to solve the proposed model, which expands the range of tabu search by changing the neighborhood structure [27,28].

### 4.1. Initial Solution

The generation of the initial solution mainly includes the following three stages:

(1) Generation of pure-truck delivery route

The generation of pure-truck delivery routes uses the nearest neighbor algorithm, and the steps are as follows:

Step 1. Create an empty route $R_n$ for each truck, $R_n = \varnothing, \forall n \in N$.

Step 2. Assume that the last customer point on route $R_n$ is $g$, select a customer point closest to $g$ from all unassigned customer points and insert it into route $R_n$.

Step 3. Check whether there are unassigned customer points or not. If yes, go to step 2; otherwise, terminate the algorithm and output the final delivery route $\{R_n\}$.

(2)    Generation of drone delivery route

For any two customer points $i$ and $j$ on route $R_n$, if customer point $k$ satisfies: $t_L + t_R + t_{ik}^D + t_{kj}^D + s^D \leq e$, $i \rightarrow k \rightarrow j$ is a feasible drone delivery route $R_n'$, and customer point $k$ is visited by the drone.

(3)    Route update

It is assumed that $i \rightarrow k \rightarrow j$ is a pure-truck delivery route. If customer point $k$ can be visited by the drone, $k$ is removed from the route, i.e., the truck delivery route $R_n$ is updated as $i \rightarrow j$ and the drone delivery route is $R_n'$: $i \rightarrow k \rightarrow j$. The route update process is shown in Figure 2.

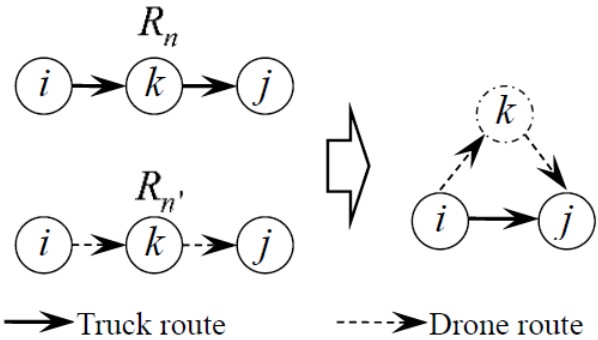

**Figure 2.** Route update.

*4.2. Variable Neighborhood Structure Design*

Expand the search range by changing the neighborhood structure so as to make it easier for the algorithm to obtain the global optimal solution. In this work, the following two operations are used to change the neighborhood structure.

(1)    Point exchange.

Select any two customer points $i$ and $j$, $i, j \in C$ in the current truck–drone route and exchange their positions until every single customer point is involved with a point exchange. The objective function value is calculated for the truck–drone route generated by each exchange. Conduct the aspiration criterion and update the tabu list.

(2)    Link exchange.

Randomly select a customer point $i$, $i \in C$ in the current truck–drone route, and separate out a part that contains customer point $i$ as a sub-route $W_i$ from the truck–drone route. Then select another customer point $j$. If customer points $i$ and $j$ are on the same truck–drone route and are not adjacent, reverse the link between $i$ and $j$; If customer points $i$ and $j$ are not on the same truck–drone route, separate out a part that contains customer point $j$ as a sub-route $W_j$. Then, exchange part of the links on $W_i$ and $W_j$ to connect customer points $i$ and $j$. Calculate the objective function value for the truck–drone route generated by each exchange. Conduct the aspiration criterion and update the tabu list.

*4.3. Tabu List and Termination Conditions*

The tabu list $T_L$ is updated according to the first-in-first-out principle. When the neighborhood moves, its reverse movement is added to the bottom of $T_L$, the earliest

movement in $T_L$ is removed from the top of $T_L$, and all moves in $T_L$ are prohibited. However, if the prohibited movement can produce a better solution than the current optimal solution, the movement will be accepted. The length of $T_L$ is set as $\left\lceil \sqrt{m} \right\rceil$.

We adopt two termination criteria: (1) The number of iterations to continuously obtain the same feasible solution is greater than $H_1$; (2) The total number of iterations reaches an upper limit value $H_2$. When any termination condition is reached, the algorithm terminates and outputs the optimal solution.

### 4.4. Algorithm Flow

The main steps of the variable neighborhood tabu search algorithm are as follows:

Step 1. Let the number of iterations be $K = 0$. Randomly generate an initial feasible solution $\Omega^0$, and calculate the initial objective function value $\tau^0$ according to $\Omega^0$ using in Equation (1). Initialize the tabu list $T_L$. Set both the current solution $\Omega^K$ and the optimal solution $\Omega^B$ as $\Omega^0$, i.e., $\Omega^K = \Omega^B = \Omega^0$, and set the current objective function value $\tau^K$ and the optimal objective function value $\tau^B$ as $\tau^0$, i.e., $\tau^K = \tau^B = \tau^0$. The number of iterations to obtain the feasible solution $\Omega^K$ is $X$.

Step 2. Generate a neighborhood solution $\Omega^N$ of $\Omega^K$, and obtain its objective function value $\tau^N$. Determine whether or not the movement operation for generating the neighborhood solution is in $T_L$. If yes, go to step 4; otherwise, go to step 3.

Step 3. Set $\Omega^K = \Omega^N$, $\tau^K = \tau^N$. Update $T_L$. If $\tau^K > \tau^N$, then set $\Omega^B = \Omega^K$, $\tau^B = \tau^K$, and go to step 4, otherwise, go to step 5.

Step 4. If $\tau^N > \tau^B$, cancel the tabu status, let $\Omega^K = \Omega^B = \Omega^N$, $\tau^K = \tau^B = \tau^N$, $K = K + 1$, and update $T_L$, go to step 5, otherwise, go to step 2.

Step 5. If $\Omega^{K+1} = \Omega^K$, let $X = X + 1$. Judge whether $K \geq H_2$ or $X \geq H_1$ is true; if yes, go to step 6; otherwise, go to step 2.

Step 6. Output the optimal solutions $\Omega^*$, $\Omega^* = \Omega^B$, and the corresponding optimal objective function value is $\tau^* = \tau^B$.

## 5. Numerical Experiment

### 5.1. Testing Delivery Network

We adopted a delivery network with 24 nodes, i.e., 1 warehouse and 23 customer points as an example to determine the optimal-delivery truck–drone route using the proposed model and algorithm, assuming that at the initial time $u_0 = 5$ min, three trucks delivered parcels to 23 customer points from a warehouse. Each truck carried a drone, and they jointly carried out the delivery process. Each customer point only received one parcel. The spatial layout of warehouse and customer points is shown in Figure 3. The average speed of trucks was $V^T = 40$ km/h, and the average speed of drones was $V^D = 60$ km/h. The distance between any two nodes is shown in Table 1, where 0 in the first row and the first column represents the warehouse ID, 1~23 represent the customer point ID, and the unit of distance is the kilometer.

In the experiment, we used rotary-wing drones. The axle base was 450 mm, the dead weight was 3200 g, the maximum load was 3000 g, the maximum endurance was 60 min, the maximum cruising distance was 60 km, the maximum flight altitude was 2000 m, and the size was 55 cm × 65 cm × 32 cm.

The trucks used in this experiment were van trucks. The wheelbase was 1360 mm, the dead weight was 1.275 ton, the load capacity was 0.8 ton, the maximum horsepower 116 kw, the torque was 150 N·m, and the size was 4.915 m × 1.63 m × 2.475 m.

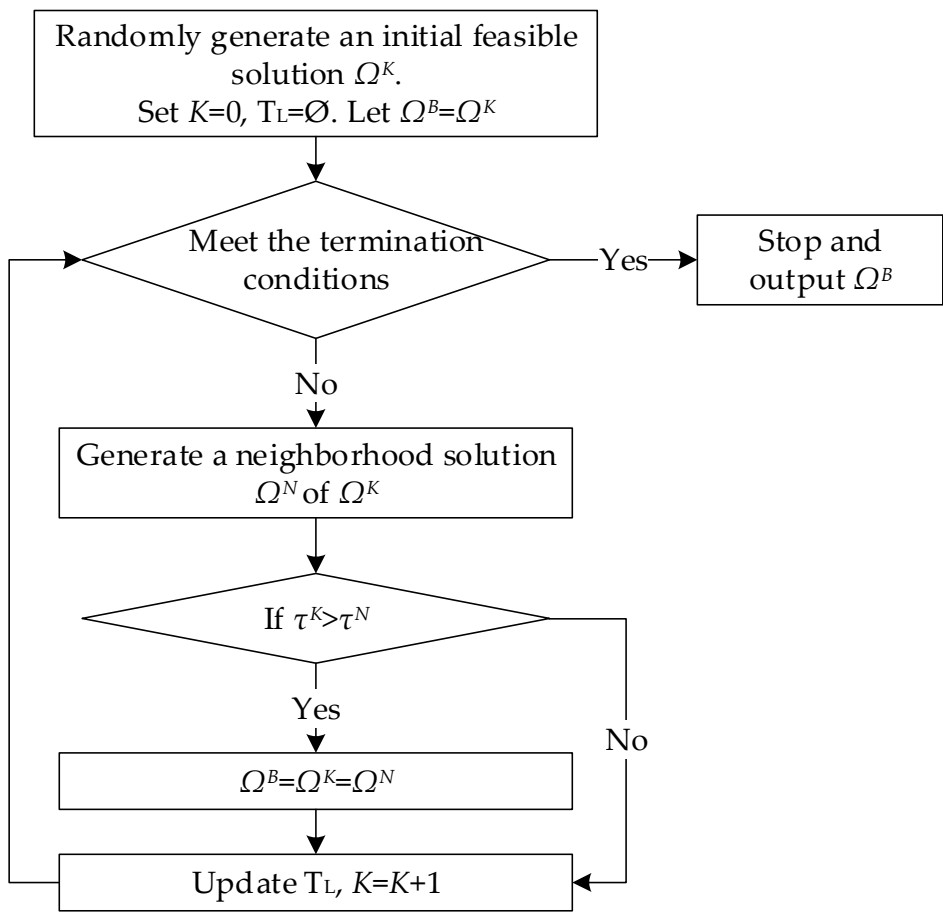

**Figure 3.** Flow chart of tabu search algorithm.

**Table 1.** Distance between nodes.

| Nodes | 0 | 1 | 2 | 3 | 4 | 5 | 6 | 7 | 8 | 9 | 10 | 11 | 12 | 13 | 14 | 15 | 16 | 17 | 18 | 19 | 20 | 21 | 22 | 23 |
|---|---|---|---|---|---|---|---|---|---|---|---|---|---|---|---|---|---|---|---|---|---|---|---|---|
| 0 | 0 | 15 | 16 | 13 | 16 | 14 | 18 | 21 | 31 | 29 | 30 | 19 | 26 | 21 | 20 | 29 | 35 | 37 | 36 | 42 | 41 | 39 | 35 | 37 |
| 1 | 15 | 0 | 17 | 19 | 31 | 32 | 14 | 13 | 16 | 15 | 33 | 33 | 36 | 37 | 32 | 33 | 22 | 23 | 28 | 38 | 39 | 37 | 37 | 41 |
| 2 | 16 | 17 | 0 | 7 | 28 | 26 | 29 | 27 | 23 | 10 | 18 | 13 | 28 | 30 | 41 | 45 | 23 | 15 | 25 | 34 | 39 | 40 | 41 | 44 |
| 3 | 13 | 19 | 7 | 0 | 25 | 26 | 29 | 29 | 34 | 18 | 11 | 14 | 18 | 20 | 32 | 43 | 31 | 18 | 29 | 34 | 35 | 35 | 42 | 46 |
| 4 | 16 | 31 | 28 | 25 | 0 | 16 | 28 | 31 | 44 | 35 | 32 | 30 | 20 | 15 | 23 | 36 | 47 | 41 | 38 | 43 | 36 | 19 | 14 | 32 |
| 5 | 14 | 32 | 26 | 26 | 16 | 0 | 29 | 34 | 42 | 39 | 37 | 36 | 36 | 33 | 15 | 29 | 45 | 46 | 42 | 45 | 43 | 36 | 20 | 21 |
| 6 | 18 | 14 | 29 | 29 | 28 | 29 | 0 | 9 | 29 | 32 | 39 | 38 | 40 | 40 | 19 | 12 | 36 | 44 | 47 | 51 | 51 | 48 | 43 | 41 |
| 7 | 21 | 13 | 27 | 29 | 31 | 34 | 9 | 0 | 20 | 22 | 39 | 39 | 41 | 42 | 25 | 23 | 21 | 39 | 45 | 50 | 49 | 48 | 45 | 47 |
| 8 | 31 | 16 | 23 | 34 | 44 | 42 | 29 | 20 | 0 | 21 | 35 | 37 | 40 | 41 | 38 | 37 | 16 | 35 | 45 | 52 | 53 | 52 | 50 | 53 |
| 9 | 29 | 15 | 10 | 18 | 35 | 39 | 32 | 22 | 21 | 0 | 19 | 21 | 26 | 29 | 41 | 43 | 16 | 20 | 24 | 41 | 43 | 43 | 45 | 50 |
| 10 | 30 | 33 | 18 | 11 | 32 | 37 | 39 | 39 | 35 | 19 | 0 | 9 | 17 | 20 | 48 | 52 | 34 | 15 | 16 | 26 | 30 | 32 | 35 | 53 |
| 11 | 19 | 33 | 13 | 14 | 30 | 36 | 38 | 39 | 37 | 21 | 9 | 0 | 15 | 17 | 48 | 50 | 46 | 19 | 14 | 23 | 24 | 25 | 37 | 48 |
| 12 | 26 | 36 | 28 | 18 | 20 | 36 | 40 | 41 | 40 | 26 | 17 | 15 | 0 | 6 | 47 | 52 | 52 | 34 | 15 | 21 | 22 | 22 | 34 | 47 |
| 13 | 21 | 37 | 30 | 20 | 15 | 33 | 40 | 42 | 41 | 29 | 20 | 17 | 6 | 0 | 46 | 52 | 54 | 43 | 24 | 29 | 20 | 13 | 21 | 32 |
| 14 | 20 | 32 | 41 | 32 | 23 | 15 | 19 | 25 | 38 | 41 | 48 | 48 | 47 | 46 | 0 | 16 | 50 | 52 | 51 | 56 | 55 | 51 | 37 | 19 |
| 15 | 29 | 33 | 45 | 43 | 36 | 29 | 12 | 23 | 37 | 43 | 52 | 50 | 52 | 52 | 16 | 0 | 49 | 53 | 56 | 60 | 58 | 49 | 45 | 31 |
| 16 | 35 | 22 | 23 | 31 | 47 | 45 | 36 | 21 | 16 | 16 | 34 | 46 | 52 | 54 | 50 | 49 | 0 | 31 | 36 | 51 | 52 | 53 | 53 | 56 |
| 17 | 37 | 23 | 15 | 18 | 41 | 46 | 44 | 39 | 35 | 20 | 15 | 19 | 34 | 43 | 52 | 53 | 31 | 0 | 31 | 36 | 40 | 41 | 43 | 48 |
| 18 | 36 | 28 | 25 | 29 | 38 | 42 | 47 | 45 | 45 | 24 | 16 | 14 | 15 | 24 | 51 | 56 | 36 | 31 | 0 | 16 | 22 | 29 | 35 | 51 |
| 19 | 42 | 38 | 34 | 34 | 43 | 45 | 51 | 50 | 52 | 41 | 26 | 23 | 21 | 29 | 56 | 60 | 51 | 36 | 16 | 0 | 15 | 32 | 38 | 45 |
| 20 | 41 | 39 | 39 | 35 | 36 | 43 | 51 | 49 | 53 | 43 | 30 | 24 | 22 | 20 | 55 | 58 | 52 | 40 | 22 | 15 | 0 | 16 | 29 | 42 |
| 21 | 39 | 37 | 40 | 35 | 19 | 36 | 48 | 48 | 52 | 43 | 32 | 25 | 22 | 13 | 51 | 49 | 53 | 41 | 29 | 32 | 16 | 0 | 15 | 38 |
| 22 | 35 | 37 | 41 | 42 | 14 | 20 | 43 | 45 | 50 | 45 | 35 | 37 | 34 | 21 | 37 | 45 | 53 | 43 | 35 | 38 | 29 | 15 | 0 | 21 |
| 23 | 37 | 41 | 44 | 46 | 32 | 21 | 41 | 47 | 53 | 50 | 53 | 48 | 47 | 32 | 19 | 31 | 56 | 48 | 51 | 45 | 42 | 38 | 21 | 0 |

### 5.2. Results

We set (1) model parameters: $M = 10{,}000$, $t_L = 1$ min, $t_R = 1$ min, $s^T = 2$ min, $s^D = 2$ min, $e = 50$ min; (2) algorithm parameters: $H_1 = 100$, $H_2 = 1000$. The procedure is coded in MATLAB R2020a, and all experiments are conducted on a Windows Server 2019 server with an Intel Core i9-10900X CPU (4.7 GHZ) and 128 GB DDR4 RAM.

Table 2 demonstrates three delivery routes of the three truck–drone teams, including $u_j^n$, $u_j^{n'}$, $z_i^n$, $\pi_i^n$, and $\tau$. The unit of $u_j^n$, $u_j^{n'}$, $\pi_i^n$ and $\tau$ is the minute.

**Table 2.** Optimal delivery route and total delivery time of each route.

| Route Code | | | | | Optimal Delivery Route | | | | | | | $\tau$ |
|---|---|---|---|---|---|---|---|---|---|---|---|---|
| A | $i, z_i^n$ | 0 | 7 | 1 | 8 | 16 | 9 | 17 | 10 | 2 | 0 | 187 |
| | $u_j^n$ | 5 | - | 28 | - | 76 | - | 127 | - | 153 | 192 | |
| | $u_j^{n'}$ | 5 | 26 | 41 | 59 | 77 | 95 | 117 | 146 | 166 | 192 | |
| | $\pi_i^n$ | - | - | 13 | - | 1 | - | 0 | - | 13 | - | |
| B | $i, z_i^n$ | 0 | 3 | 11 | 12 | 18 | 19 | 20 | 21 | 13 | 0 | 167 |
| | $u_j^n$ | 5 | - | 34 | - | 59 | - | 103 | - | 137 | 172 | |
| | $u_j^{n'}$ | 5 | 18 | 34 | 51 | 68 | 86 | 103 | 121 | 136 | 172 | |
| | $\pi_i^n$ | - | - | 0 | - | 10 | - | 0 | - | 0 | - | |
| C | $i, z_i^n$ | 0 | 5 | 4 | 22 | 23 | 14 | 15 | 6 | 0 | - | 172 |
| | $u_j^n$ | 5 | - | 29 | - | 87 | - | 140 | - | 177 | - | |
| | $u_j^{n'}$ | 5 | 19 | 37 | 53 | 80 | 101 | 119 | 154 | 176 | - | |
| | $\pi_i^n$ | - | - | 8 | - | 0 | - | 0 | - | - | - | |

On route A, the truck and drone jointly served eight customer points. The truck route was 0→1→16→17→2→0, i.e., the truck served four customer points. The drone's routes were 0→7→1, 1→8→16, 16→9→17, 17→10→2. After the drone finished the last delivery task at customer point 10, it was collected by the truck at customer point 2, then the truck carried the drone back to the warehouse. The delivery process took 187 min.

On route B, the truck and drone also visited eight customer points. The truck route was 0→11→18→20→13→0. The drone routes included 0→3→11, 11→12→18, 18→19→20, 20→21→13, i.e., the drone visited four customer points. The drone returned to the truck at customer point 13 after customer point 21 was visited. The total delivery time was 167 min.

On route C, the truck route was 0→4→23→15→0, i.e., the truck delivered parcels to three customer points. The drone flew along three routes: 0→5→4, 4→22→23, 23→14→15, 15→6→0. The drone and the truck returned to the warehouse separately, and the drone arrived at the warehouse 1 min earlier than the truck. The total delivery time was 172 min.

According to Equation (1), the total delivery time $\tau$ was $max\{187, 167, 172\} = 187$ min. Figure 4 illustrates the three optimal truck–drone delivery routes.

As shown in Figure 5, on the above three routes, truck–drone delivery can save 47 min, 48 min and 38 min respectively compared with pure-truck delivery. The whole delivery process can save up to 20.1% of the time.

Obviously, truck–drone delivery can save total delivery time, but delivery enterprises must purchase drones, which is a large investment (or cost). In reality, there is a possibility that the benefits of saving delivery time do not offset this cost. Hence, the tradeoff between saving delivery time and purchasing drones should be considered when enterprises determine whether to implement truck–drone delivery.

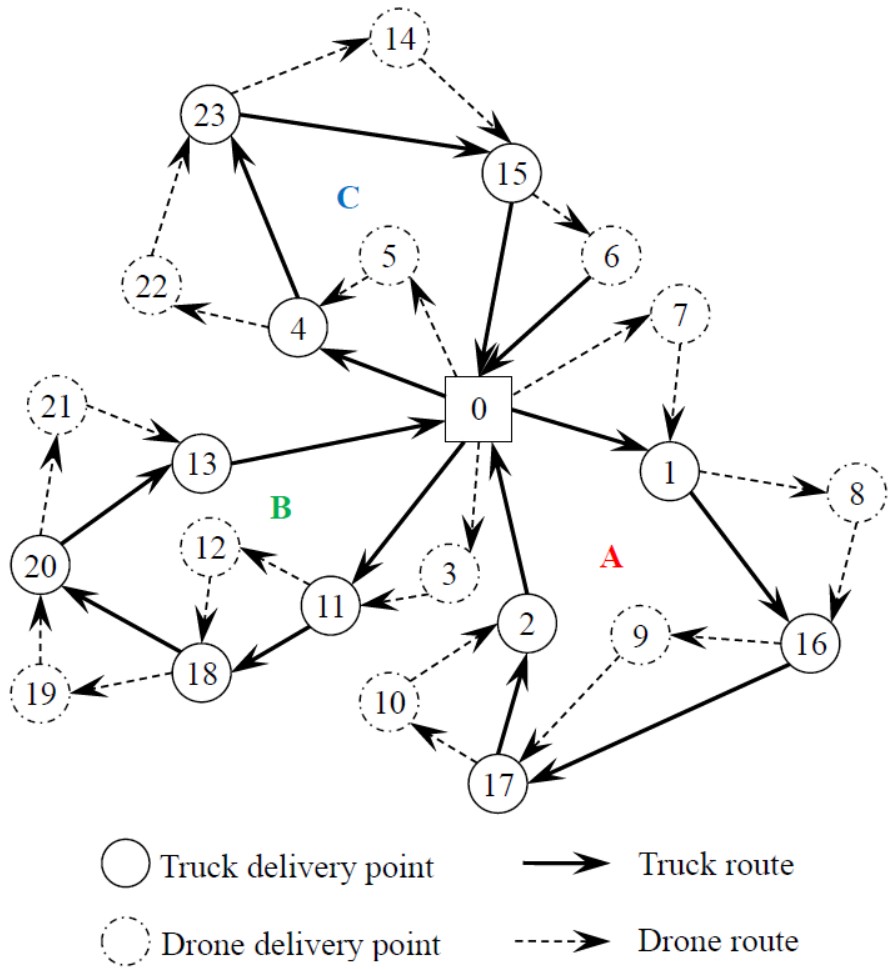

**Figure 4.** Optimal truck–drone delivery routes.

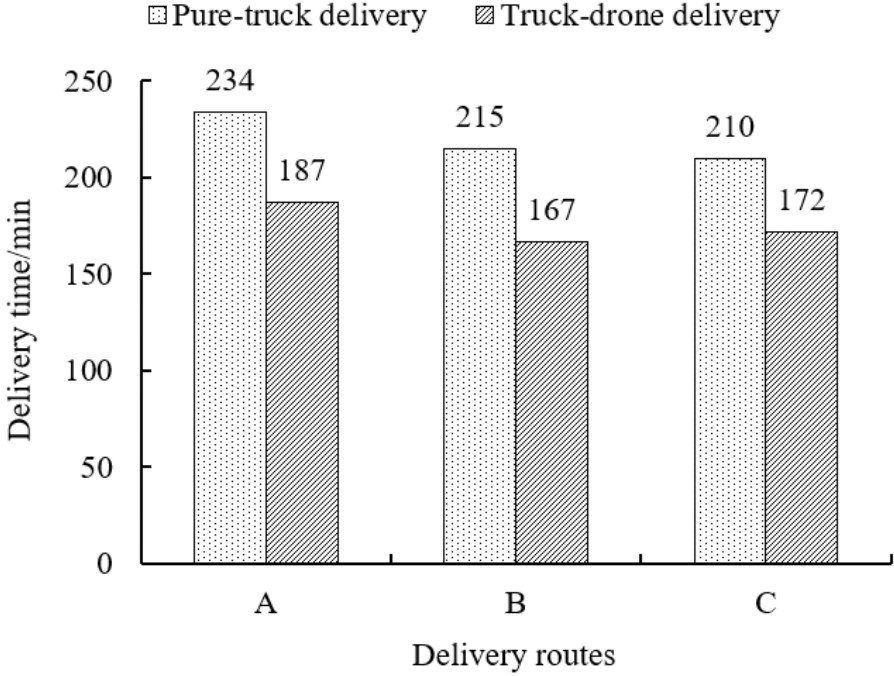

**Figure 5.** Comparison of truck–drone delivery and pure-truck delivery.

### 5.3. Sensitivity Analysis

This section mainly discusses the effects of the number of truck–drones, truck speed and drone speed on the total delivery time.

#### 5.3.1. Number of Truck–drones

The number of truck–drones $v$ is directly related to the total delivery time $\tau$. It assumes that the $v$ varies from 1 to 4, and the sensitivity analysis results of $v$ are shown in Figure 6. With the increase in $v$, $\tau$ shows an obviously decreasing trend, but the decreasing speed of $\tau$ gradually slows down, indicating that the marginal benefit generated by increasing $v$ is gradually reduced; that is, after $v$ reaches a certain amount, continuing to increase $v$ will not significantly reduce $\tau$.

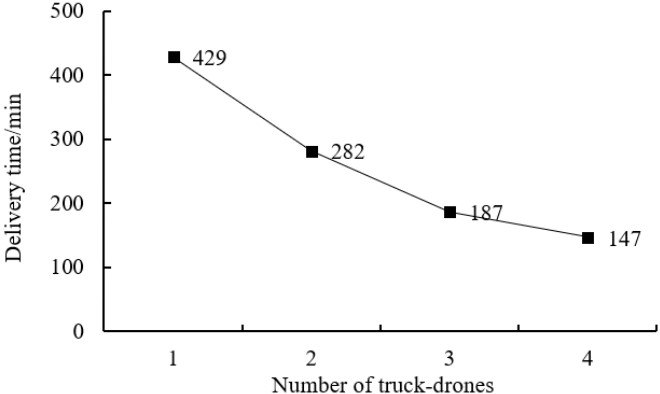

**Figure 6.** Sensitivity analysis of the number of truck–drones.

#### 5.3.2. Truck Speed and Drone Speed

Set the speed growth rates of trucks and drones respectively as $\rho^T$ and $\rho^D$, and both $\rho^T$ and $\rho^D$ increase from $-20\%$ to $20\%$ by $5\%$. The sensitivity analysis results are shown in Figure 7. Only increasing $\rho^T$ or $\rho^D$ can reduce $\tau$. The main reason is that the increase in $\rho^T$ or $\rho^D$ can shorten the time trucks wait for drones or drones wait for trucks at rendezvous points. However, just increasing either $\rho^T$ or $\rho^D$ has only a small effect on reducing $\tau$, and the decreasing rate $\tau$ slows down as $\rho^T$ or $\rho^D$ increases. In addition, only increasing either $\rho^T$ or $\rho^D$ cannot obtain minimum $\tau$. Only when both $\rho^T$ and $\rho^D$ are $20\%$ can we obtain minimum $\tau$, which is 162 min. It shows that only when the truck speed and drone speed increase simultaneously can the delivery time be significantly reduced. Our result is in line with the conclusion given by Carlsson and Song [29].

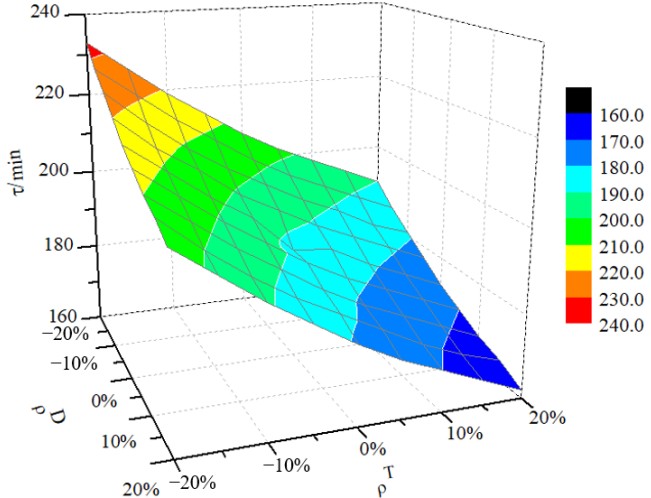

**Figure 7.** Sensitivity analysis of truck speed and drone speed.

## 6. Conclusions

This study focused on the optimal route problem for truck–drone delivery, which is a classic FSTSP. We considered the scenario that the drones may arrive at rendezvous points later than the trucks. This problem was formulated as a single-objective mixed-integer nonlinear programming model subject to time constraints and route constraints, aiming at minimization of total delivery time. Then, we developed a variable neighborhood tabu search algorithm to solve the model. Finally, a testing delivery network was used to illustrate the proposed method. The optimal truck–drone delivery route included the sequence of all delivery tasks, and the total delivery time. The key findings are as follows:

(1) Compared with traditional truck delivery, truck–drone delivery can save delivery time by 20.1% in the numerical experiment, which can guide logistics enterprises to complete the delivery tasks in less time.

(2) More trucks and drones can effectively reduce the total delivery time, but the marginal benefit gradually decreases. Only increasing either the truck speed or drone speed cannot minimize the delivery time, and only when the truck speed and drone speed increase synchronously can the delivery time be significantly reduced.

The results of the numerical experiment indicate that the proposed method can generate the optimal route for truck–drone delivery, taking truck waiting time at rendezvous points into consideration, which provides a reference for logistics enterprises in planning delivery routes. However, there are still some limitations of this study. Future work will focus on: (1) taking uncertainties, e.g., truck and drone speed and service time at customer points into account during the entire delivery time; (2) designing more heuristic algorithms and comparing these algorithms to find a more efficient model solution.

**Author Contributions:** B.T. wrote the manuscript; J.W. collected the data; X.W. edited and revised the manuscript; F.Z. drew the figures; X.M. designed research methods; W.Z. analyzed the data. All authors have read and agreed to the published version of the manuscript.

**Funding:** This work was supported by the Fundamental Research Funds for the Central Universities (Grant Number 300102341513) and the Natural Science Research Program of Shaanxi Province (Grant Number 2020JQ-360) and the National Natural Science Foundation of China (Grant Number 52102374) and the Transportation Science and Technology Research Project of Hebei Province (Grant Number JX-202006).

**Institutional Review Board Statement:** Not applicable.

**Informed Consent Statement:** Not applicable.

**Data Availability Statement:** Data are contained within the article. The data presented in this study can be requested from the authors.

**Conflicts of Interest:** The authors declare no conflict of interest.

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
