# Peer review of "Optimal Route Planning for Truck–Drone Delivery Using Variable Neighborhood Tabu Search Algorithm"

_applsci, doi:10.3390/app12010529_

Round 1

Reviewer 1 Report

This paper tackles the problem of optimally defining the delivery route for a truck-drone delivery system. To this aim, the authors propose the definition of a mixed-integer non-linear programming model and use a variable neighborhood tabu search algorithm for its resolution. In my opinion, the main novelty of the work consists in considering the truck waiting time for the arrival of the drone at rendezvous points when formulating the problem.

The topic of the article and the discussed application are particularly interesting and actual. Its resonance is relevant in the context of logistics and in particular with respect to its evolution into Logistics 4.0.

The state of the art requires to be revised providing a more critical discussion of the reported documents. In its present form, it is more a list of descriptions of the existing papers in the field instead of being a critical review of the state of the art. Moreover, this section is also missing the positioning of the paper within the state of the art. Authors should compare the presented paper with the existing works highlighting the novelty and the advantages of the proposed approach, both in terms of practical application and theoretical/methodological contribution. Further recent papers on this topic can be included such as 10.1109/MED51440.2021.9480359.

The numerical experiments section should include more detailed technical specifications of the considered trucks and drones for the described system. Moreover, the results should also consider a discussion of the economical benefits for the company in implementing the proposed solution.  In fact, the authors state that a reduction of transportation time can be obtained by increasing 20% the speed of the drone and the truck. However, this might translate into a huge increase in the purchase price of the drone that satisfies the speed specifications and thus the reduction of transportation time becomes in fact not beneficial for the company.

Author Response

Point 1: This paper tackles the problem of optimally defining the delivery route for a truck-drone delivery system. To this aim, the authors propose the definition of a mixed-integer non-linear programming model and use a variable neighborhood tabu search algorithm for its resolution. In my opinion, the main novelty of the work consists in considering the truck waiting time for the arrival of the drone at rendezvous points when formulating the problem.

The topic of the article and the discussed application are particularly interesting and actual. Its resonance is relevant in the context of logistics and in particular with respect to its evolution into Logistics 4.0.

Response 1: The authors greatly appreciate the reviewer’s encouragement and suggestions. The authors have revised the manuscript according to the reviewer’s comments.

Point 2: The state of the art requires to be revised providing a more critical discussion of the reported documents. In its present form, it is more a list of descriptions of the existing papers in the field instead of being a critical review of the state of the art. Moreover, this section is also missing the positioning of the paper within the state of the art. Authors should compare the presented paper with the existing works highlighting the novelty and the advantages of the proposed approach, both in terms of practical application and theoretical/methodological contribution. Further recent papers on this topic can be included such as 10.1109/MED51440.2021.9480359.

Response 2: The authors have highlighted the novelty and the advantages of this paper. Please refer to lines 116-120.

Point 3: The numerical experiments section should include more detailed technical specifications of the considered trucks and drones for the described system. Moreover, the results should also consider a discussion of the economical benefits for the company in implementing the proposed solution.  In fact, the authors state that a reduction of transportation time can be obtained by increasing 20% the speed of the drone and the truck. However, this might translate into a huge increase in the purchase price of the drone that satisfies the speed specifications and thus the reduction of transportation time becomes in fact not beneficial for the company.

Response 3: The authors agree with the review’s comment. The economical benefits for the company will be considered in our future research.

Reviewer 2 Report

This paper formulate the optimal delivery route problem for truck-drone delivery using a mixed integer nonlinear programming model subject to time constraints and route constraints aiming to minimize the total delivery time. The model is properly formulated the results and is evaluated using simple benchmark data.

In Reference No11, the title of this paper is missing.

I recommend you to include the following two papers for your reference.
John Gunnar Carlsson, Siyuan Song (2017) Coordinated Logistics with a Truck and a Drone. Management Science 64(9):4052-4069.
Stefan Poikonen, James F. Campbell(2021) Future directions in drone routing research, Networks, Volume77, Issue1, 116-126.

Author Response

Point 1: This paper formulate the optimal delivery route problem for truck-drone delivery using a mixed integer nonlinear programming model subject to time constraints and route constraints aiming to minimize the total delivery time. The model is properly formulated the results and is evaluated using simple benchmark data.

Response 1: The authors greatly appreciate the reviewer’s encouragement and suggestions. The authors have revised the manuscript according to the reviewer’s comments.

Point 2: In Reference No11, the title of this paper is missing.

Response 2: The authors have added the title into Reference No11.

Point 3: I recommend you to include the following two papers for your reference.

John Gunnar Carlsson, Siyuan Song (2017) Coordinated Logistics with a Truck and a Drone. Management Science 64(9):4052-4069.

Stefan Poikonen, James F. Campbell(2021) Future directions in drone routing research, Networks, Volume77, Issue1, 116-126.

Response 3: The authors appreciate the reviewer’s suggestion. These two papers help a lot in improving the literature review in our paper. Please refer to Reference No 3 and 28.

Reviewer 3 Report

The paper focuses on the optimal route problem for truck-drone delivery. They formulate the problem as a single-objective mixed integer nonlinear programming model subject to time constraints and route constraints aiming at the minimization of total delivery time. In order to solve the problem Authors propose an interesting variable neighborhood tabu search algorithm. Finally, they demonstrate the proposed model and algorithm on a delivery network and analyze the effects of different model parameters on the delivery efficiency.

The paper is quite interesting and well written. Its construction is correct with main elements of such papers: introduction, literature review, definition of the problem, proposal of the model and methods to solve it, and computational experiment with main conclusions. In my opinion, the most interesting part is definition of the model and proposal of the algorithm based od VNS and TS. It is quite interesting and original but in order to improve readability of the paper I suggest to rewrite secion 4.4 Algorithm Flow using the form of algorithm instead of list of steps (for example similar to latex \algorithm \algorithmics environment).

Computational experiment has been carried out correctly allowing the Authors to obtain many interesting results, for example:  compared with the traditional truck delivery, the truck-drone delivery can save delivery time by 20.1% in the numerical experiment and that more trucks and drones can effectively reduce the total delivery time, but the marginal benefit gradually decreases. 

Although Authors' conclusions are interesting and important, in my opinion it is hard to generalized obtained results. Authors use only a single instance with 24 nodes (rather small instance) and it is not clear how this instance has been obtained (is it instance based on real problem?). As I can suppose, it is also hard to compare obtained results with other Authors' methods. Isn't it?

I suggest to extend the computational experiment.

I have also a comment about greedy random adaptive search algorithm (GRASA) - page 3. In most papers and books referring to (meta)heuristics a form greedy random adaptive search prodcedure (GRASP) is rather prefered. Originaly, P. Moscato also proposed this method as GRASP.

In my opinion the paper merits publication in the Applied Science journal, but it requires a few improvements.

Author Response

Point 1: The paper focuses on the optimal route problem for truck-drone delivery. They formulate the problem as a single-objective mixed integer nonlinear programming model subject to time constraints and route constraints aiming at the minimization of total delivery time. In order to solve the problem Authors propose an interesting variable neighborhood tabu search algorithm. Finally, they demonstrate the proposed model and algorithm on a delivery network and analyze the effects of different model parameters on the delivery efficiency.

The paper is quite interesting and well written. Its construction is correct with main elements of such papers: introduction, literature review, definition of the problem, proposal of the model and methods to solve it, and computational experiment with main conclusions. In my opinion, the most interesting part is definition of the model and proposal of the algorithm based od VNS and TS. It is quite interesting and original.

Response 1: The authors greatly appreciate the reviewer’s encouragement. The authors have revised the manuscript according to the reviewer’s comments.

Point 2: in order to improve readability of the paper, I suggest to rewrite section 4.4 Algorithm Flow using the form of algorithm instead of list of steps (for example similar to latex \algorithm \algorithmics environment).

Response 2: In order to improve readability of this paper, the authors added a flow chart of tabu search algorithm. Please refer to Figure 3.

Point 3: Computational experiment has been carried out correctly allowing the Authors to obtain many interesting results, for example:  compared with the traditional truck delivery, the truck-drone delivery can save delivery time by 20.1% in the numerical experiment and that more trucks and drones can effectively reduce the total delivery time, but the marginal benefit gradually decreases.

Response 3: The authors greatly appreciate the reviewer’s encouragement.

Point 4: Although Authors' conclusions are interesting and important, in my opinion it is hard to generalized obtained results. Authors use only a single instance with 24 nodes (rather small instance) and it is not clear how this instance has been obtained (is it instance based on real problem?). As I can suppose, it is also hard to compare obtained results with other Authors' methods. Isn't it?

I suggest to extend the computational experiment.

Response 4: The authors appreciate the reviewer’s suggestion. The layout of delivery network used in the paper is extracted from a real delivery network. Because of business secret, we do not provide all information of the network. In the future research, we will demonstrate our method on a more complicated delivery network.

Point 5: I have also a comment about greedy random adaptive search algorithm (GRASA) - page 3. In most papers and books referring to (meta)heuristics a form greedy random adaptive search prodcedure (GRASP) is rather prefered. Originaly, P. Moscato also proposed this method as GRASP.

Response 5: The authors agree with the review’s comment.

Reviewer 4 Report

Interesting work. However, a few things need to be adjusted:
1. All abbreviations mentioned for the first time must have their full names.
2. View English language and concepts (e.g. "... large body of studies" etc .....
3. It is not very clear why you choose the first category in the introduction ("This study falls in the first category")
3. The article must be prepared in accordance with the structural requirements (Introduction, Methods and Methodology, Results and Discussion, Conclusions). Because under the current structure, I would not agree that Chapter 3 is a formulation of the problem. Rather, it is a section describing the methodology.
4. Move Figure 1 closer to the reference text. However, with regard to Figure 1, it is not clear why it is based on this principle. Because if you rely on the traveling salesman method, it should be more like a circular route.
5. The title of Chapter 4 should be clarified as it is not clear what type of model.
6. Table 1 must fit on one sheet. It must also be made clear which units of measurement are used. Also in the text, perhaps at item 23 to explain what the numbers 0 to 23 mean in the table (or the need for 0 not to be written at all in the Nodes)
7. The title of Table 2 should be clarified, as it is not clear what the results of these calculations are (volume, velocity or what?).
8. The statements in the conclusions raise additional questions, so in my view that information should be incorporated into the description of the methodology and conditions (e.g. you write “.... may arrive at rendezyous points later than trucks). Hence the question - and whether the weather conditions were assessed here, how they would affect the result. It is also unclear whether the amount of cargo transported by the drone and truck is the same and so on.

Author Response

Point 1: Interesting work. However, a few things need to be adjusted:

Response 1: The authors greatly appreciate the reviewer’s encouragement. The authors have revised the manuscript according to the reviewer’s comments.

Point 2: All abbreviations mentioned for the first time must have their full names.

Response 2: The authors have double checked all abbreviations in the manuscript and provide the full names of “NP-hard”. Please refer to lines 17-18, and line 96. Additionally, the authors have changed “UAVs” in line 294 into “drones”.

Point 3: View English language and concepts (e.g. "... large body of studies" etc .....

Response 3: The authors have changed “a large body of studies” into “a lot of studies”.

Point 4: It is not very clear why you choose the first category in the introduction ("This study falls in the first category").

Response 4: The authors have changed “This study falls in the first category” into “This study mainly studies FSTSP”, which just a description of the research problem in this paper.

Point 5: The article must be prepared in accordance with the structural requirements (Introduction, Methods and Methodology, Results and Discussion, Conclusions). Because under the current structure, I would not agree that Chapter 3 is a formulation of the problem. Rather, it is a section describing the methodology.

Response 5: The authors have changed the title of Section 3 into “Methodology”.

Point 6: Move Figure 1 closer to the reference text. However, with regard to Figure 1, it is not clear why it is based on this principle. Because if you rely on the traveling salesman method, it should be more like a circular route.

Response 6: The authors have moved Figure 1 closer to the reference text. Actually, trucks depart from a warehouse and return to the warehouse. For simplicity, this paper describes this delivery process as “the truck departs from {0} and returns to {m+1}”. Both 0 and m+1 represent the warehouse. Absolutely, the route is a circular route. We cut the circle and expand it into a linear route showed in Figure 1.

Point 7: The title of Chapter 4 should be clarified as it is not clear what type of model.

Response 7: The authors have clarified the type of model in lines 169-170, which is a single objective mixed integer nonlinear programming model.

Point 8: Table 1 must fit on one sheet. It must also be made clear which units of measurement are used. Also in the text, perhaps at item 23 to explain what the numbers 0 to 23 mean in the table (or the need for 0 not to be written at all in the Nodes).

Response 8: The authors have clarified the units as well as the meanings of each node. Please refer to lines 301-304.

Point 9: The title of Table 2 should be clarified, as it is not clear what the results of these calculations are (volume, velocity or what?).

Response 9: The authors have changed the title of Table 2 into “Optimal delivery route and total delivery time of each route”.

Point 10: The statements in the conclusions raise additional questions, so in my view that information should be incorporated into the description of the methodology and conditions (e.g. you write “.... may arrive at rendezyous points later than trucks). Hence the question - and whether the weather conditions were assessed here, how they would affect the result. It is also unclear whether the amount of cargo transported by the drone and truck is the same and so on.

Response 10: How the weather conditions affect the results will be studied in the future research. In Section 3.1 and 3.2, we assume that trucks and drones only deliver parcels in this paper, and only one parcel is delivered to a customer.

Round 2

Reviewer 1 Report

The authors have barely addressed the previous comments and requests for improvement.

Author Response

Point 1: The authors have barely addressed the previous comments and requests for improvement.

Response 1: The authors reviewed the manuscript point by point again according to the comments in round 1.

comments in round 1

Point 1: The state of the art requires to be revised providing a more critical discussion of the reported documents. In its present form, it is more a list of descriptions of the existing papers in the field instead of being a critical review of the state of the art. Moreover, this section is also missing the positioning of the paper within the state of the art. Authors should compare the presented paper with the existing works highlighting the novelty and the advantages of the proposed approach, both in terms of practical application and theoretical/methodological contribution. Further recent papers on this topic can be included such as 10.1109/MED51440.2021.9480359.

Response 1: The authors added two paragraphs in Literature Review to summarize and compare the existing studies and highlight the novelty and the advantages of this paper. Please refer to lines 117-127.

The authors also added “10.1109/MED51440.2021.9480359” as a reference. Please see Reference 1.

Point 2: The numerical experiments section should include more detailed technical specifications of the considered trucks and drones for the described system. Moreover, the results should also consider a discussion of the economical benefits for the company in implementing the proposed solution. In fact, the authors state that a reduction of transportation time can be obtained by increasing 20% the speed of the drone and the truck. However, this might translate into a huge increase in the purchase price of the drone that satisfies the speed specifications and thus the reduction of transportation time becomes in fact not beneficial for the company.

Response 2: The authors have added technical specifications of the considered trucks and drones in Numerical Experiment. Please refer to Lines 312-318.

The authors have added a brief economical benefit analysis in Numerical Experiment. Please refer to Lines 348-352.

Reviewer 2 Report

I think the comments have been properly revised.

Author Response

Point 1: I think the comments have been properly revised.

Response 1: The authors greatly appreciate the reviewer’s encouragement and suggestions.

Reviewer 3 Report

None

Author Response

Point 1: None.

Response 1: The authors greatly appreciate the reviewer’s encouragement.
